# Genome-scale Capture C promoter interactions implicate effector genes at GWAS loci for bone mineral density

Alessandra Chesi[1], Yadav Wagley[2], Matthew E. Johnson[1], Elisabetta Manduchi[1,3], Chun Su[1], Sumei Lu[1], Michelle E. Leonard[1], Kenyaita M. Hodge[1], James A. Pippin [1], Kurt D. Hankenson [2], Andrew D. Wells[1,4] & Struan F.A. Grant [1,5,6]

Osteoporosis is a devastating disease with an essential genetic component. GWAS have discovered genetic signals robustly associated with bone mineral density (BMD), but not the precise localization of effector genes. Here, we carry out physical and direct variant to gene mapping in human mesenchymal progenitor cell-derived osteoblasts employing a massively parallel, high resolution Capture C based method in order to simultaneously characterize the genome-wide interactions of all human promoters. By intersecting our Capture C and ATAC-seq data, we observe consistent contacts between candidate causal variants and putative target gene promoters in open chromatin for ~17% of the 273 BMD loci investigated. Knockdown of two novel implicated genes, *ING3* at 'CPED1-WNT16' and *EPDR1* at 'STARD3NL', inhibits osteoblastogenesis, while promoting adipogenesis. This approach therefore aids target discovery in osteoporosis, here on the example of two relevant genes involved in the fate determination of mesenchymal progenitors, and can be applied to other common genetic diseases.

[1] Center for Spatial and Functional Genomics, Children's Hospital of Philadelphia, Philadelphia 19104 PA, USA. [2] Department of Orthopaedic Surgery, University of Michigan Medical School, Ann Arbor 48109 MI, USA. [3] Institute for Biomedical Informatics, University of Pennsylvania Perelman School of Medicine, Philadelphia 19104 PA, USA. [4] Department of Pathology and Laboratory Medicine, University of Pennsylvania Perelman School of Medicine, Philadelphia 19104 PA, USA. [5] Department of Pediatrics, University of Pennsylvania Perelman School of Medicine, Philadelphia 19104 PA, USA. [6] Divisions of Genetics and Endocrinology, Children's Hospital of Philadelphia, Philadelphia 19104 PA, USA. These authors contributed equally: Alessandra Chesi, Yadav Wagley, Matthew E. Johnson, Elisabetta Manduchi. These authors jointly supervised this work: Kurt D. Hankenson, Andrew D. Wells, Struan F.A. Grant. Correspondence and requests for materials should be addressed to S.F.A.G. (email: grants@email.chop.edu)

Osteoporosis is a common chronic form of disability due to loss of bone mineral density (BMD) and changes in bone architecture and bone material properties finally leading to an elevated fracture rate. During their lifetime, women lose 30–50% of peak bone mass, while men lose 20–30%. Fracture risk is higher in individuals with lower BMD[1–3]. Above age 50, many women of European ancestry will suffer at least one fracture; of these, many are at high risk for a subsequent fracture[4]. The subsequent loss in mobility and increased mortality have an enormous financial impact estimated at $17 billion annually[5], and this is likely to rise during the next few decades due to an aging population[4].

BMD is a classic complex trait influenced by behavioral, environmental, and genetic factors. There is strong evidence for genetic predisposition to osteoporosis[6–8], with an estimated 60–80% of the risk explained by heritable factors[9,10]. Population ancestry differences also reflect the genetic component[11,12].

After the limited successes in the candidate gene[13,14] and family-based linkage study eras of bone genetics[15,16], the GWAS approach has proven a more comprehensive and unbiased strategy to identify loci related to this complex phenotype. Increasingly higher resolution GWAS have examined adult bone phenotypes[17–22]—the latest meta-analysis reported 518 adult BMD (measured by heel ultrasound) and 13 fracture risk associated loci[23], while additional loci have been discovered in the pediatric setting[24,25]. Indeed, we found that many of these adult bone loci also operate in childhood[24–28], including rare variation at the engrailed 1 (EN1) locus[29] uncovered in a recent sequencing study in a large sample of European adults[30].

However, GWAS only reports the sentinel single nucleotide polymorphism (SNP), i.e. the SNP at a given locus with the lowest association p-value, which is unlikely to be the actual causal variant. Furthermore, the locations of the GWAS signals, the vast majority of which are non-coding in nature, residing in either intronic or intergenic regions, do not necessarily imply the precise location of the underlying effector genes. Thus, key questions related to GWAS are: is the nearest gene to a GWAS-implicated SNP in fact the actual principal culprit gene at the locus? Or is it another gene somewhere in the neighborhood? One example of this is the characterization of the key FTO locus in obesity[31–33]. The top GWAS signal resides within an intronic region of the FTO gene but in fact is primarily driving the expression of the IRX3 and IRX5 genes nearby, i.e. this variant appears to be in an embedded enhancer in one gene but influencing the expression of others. As a second example, we implicated a nearby gene, ACSL5[34], at the key type 2 diabetes TCF7L2 locus[35,36], with GTEx confirming a co-localized eQTL for this gene[37]. This should not be too surprising given that gene expression can be controlled locally or via long range interactions over large genomic distances. After all, it is well established that in non-coding regions of the genome there are important regulatory elements, such as enhancers and silencers, and genetic variants that disrupt those elements can equally confer susceptibility to complex disease. Indeed, many regulatory elements do not control the nearest genes and can reside tens or hundreds of kilobases away.

Given this, there is a compelling need to systematically characterize the mechanisms of action at each of the BMD GWAS loci, in order to identify the effector genes. Because of the paucity of public domain genomic data relevant to bone, such as eQTL and chromatin conformation capture data, we evaluate here key BMD GWAS signals[22,24,25,28,30,38,39] in the context of three-dimensional genomics by leveraging a Capture C-based high resolution promoter interactome methodology combined with ATAC-seq. The premise to utilizing a combination of Capture C and ATAC-seq is to get as close to the BMD GWAS causal variant as possible through wet lab determination in a cell line relevant for human bone biology, as opposed to statistical ascertainment—there is currently a paucity of genotyped trans-ethnic cohorts with BMD data to conduct fine mapping. We therefore carry out comprehensive ATAC-seq and Capture C in order to physically fine-map each BMD locus in primary human mesenchymal stem cell (MSC)-derived osteoblasts.

This approach can help to establish initial variant-to-gene mapping, from which therapeutic and diagnostic approaches can be developed with greater confidence given that the correct target is actually being pursued. With the need for functional insight into GWAS observations, our goal is to provide a comprehensive variant to gene mapping for BMD GWAS-implicated loci by leveraging human MSC-derived osteoblasts, a relevant cellular model for human bone biology.

## Results

**Genome-wide promoter-focused chromatin conformation capture.** In order to investigate the genome-wide contacts of all promoters in the human genome (i.e. a promoter interactome), we employed a high-resolution genome-scale, promoter-focused Capture C based approach leveraging a custom-designed SureSelect library (Agilent) targeting the transcriptional start sites (TSS) of 22,647 coding genes and 21,684 non-coding RNAs (including their alternative promoters). We adapted existing Capture C protocols[40,41] using a 4-cutter restriction enzyme (DpnII, mean fragment size 433 bp; median 264 bp) to achieve higher resolution than the more commonly used 6-cutter Hi-C related approaches[40,42] (HindIII, mean fragment size 3697 bp; median 2274 bp). The design included 36,691 bait fragments associated to 44,331 transcripts.

Primary MSCs were cultivated using standard techniques and then osteoblast induction was performed with BMP2, as previously described[43]. Nine ATAC-seq libraries derived from four donors were sequenced (Supplementary Table 1) and analyzed with the ENCODE ATAC-seq pipeline (https://github.com/kundajelab/atac_dnase_pipelines), yielding 156,406 open chromatin conservative peaks. To determine the informative genetic variants associated with BMD, we extracted 12,504 proxy SNPs for each of 110 independent signals at 107 DEXA BMD GWAS loci and 307 independent signals at 203 heel ultrasound BMD GWAS loci ($r^2 > 0.8$ to sentinel SNP in Europeans) and overlapped those variants with the positions of the open chromatin regions (ATAC-seq peaks) (Supplementary Data 1). We identified 215 informative proxy SNPs corresponding to 58 DEXA loci and 282 proxy SNPs corresponding to 112 heel loci in high LD with the sentinel SNP of the BMD loci investigated (Supplementary Data 1). This effort substantially shortened the list of candidate variants from the initial GWAS discoveries, since ATAC-seq permitted us to focus on variants residing within open chromatin regions in bone relevant cells. In order to be as inclusive as possible, so not to possibly lose any interesting lead, we also performed an analysis using a more liberal $r^2$ threshold of 0.4 (Supplementary Data 1).

Next, we performed Capture C on MSC-derived osteoblasts from three donors (Supplementary Table 1). Libraries from each donor yielded high coverage (an average of ~1.6 billion reads per library) and good quality, with >40% valid read pairs and >75% capture efficiency (% of unique valid reads captured; Supplementary Data 2). Significant interactions were called using the CHiCAGO pipeline[44]. We first performed an analysis at 1-fragment resolution and observed that single fragments had insufficient sequence depth to call longer range interactions. We therefore performed an analysis at a lower resolution (4-fragment windows, mean size ~1736 bp; median 1440 bp) to increase sensitivity at longer interaction distances. We then merged the

results from the two analyses, taking advantage of a very high resolution for short-distance interactions and trading off resolution, albeit still double the resolution of typical promoter capture Hi-C, for increased sensitivity at longer interaction distances.

Using this approach, we identified a total of 295,422 interactions (~14% were bait to bait), with a median distance for *cis* interactions of 50.5 kb, and a low number of trans-interactions (0.7%). Most of the non-bait promoter-interacting regions (PIRs) had contacts with a single baited region (84%), while only 1% contacted more than four (Supplementary Data 1). PIRs were significantly enriched for open chromatin regions detected in our ATAC-seq experiments, suggesting a potential regulatory role. They were also enriched for histone marks associated with active chromatin regions in primary human osteoblasts from the ENCODE project[45] (Supplementary Data 2), such as enhancer regions (H3K27ac and H3K4me1), active promoters (H3K4me3), actively transcribed regions (H3K36me3), transcription factor binding sites (H3K4me2); they were also enriched for CTCF binding sites and the repressive mark H3K27me3, but were depleted of the repressive mark H3K9me3 (Supplementary Figure 1A). Most interestingly, PIRs were highly enriched for BMD GWAS signals and their proxies ($r^2 > 0.4$), whereas they were not enriched for the non-bone-related Alzheimer's disease (Supplementary Figure 1 B). The number of contacts per bait were also significantly higher ($p < 2 \times 10^{-16}$, Welch's *t*-test) for open vs inaccessible promoters, as determined by ATAC-seq, but only when considering contacts to open PIRs.

To explore the relationship between promoter interactivity measured by Capture C and gene expression, we performed RNA-seq on MSC-derived osteoblasts from the same three donors. Markers of osteoblast differentiation such as *SPP1*, *SOST*, *DKK1*, *Osterix* (*SP7*), and *DMP1*, were abundantly expressed (>50th percentile) in all three samples (highlighted in red in Supplementary Data 3). We also observed an increased number of contacts in the more highly expressed genes, but only for the contacts involving both open promoters and open PIRs (Supplementary Figure 2). This suggests that combining the open chromatin landscape (ATAC-seq) with chromatin interactions data (Capture C) is an efficient approach to link biologically significant genes with their active regulatory elements.

Using this approach, we went further to map informative variants of BMD GWAS loci to their target genes. Of the BMD GWAS loci in MSC-derived osteoblasts, 46 GWAS loci revealed at least one or more BMD proxy SNPs in open chromatin (and not residing in a baited promoter region) interacting with an open gene promoter. A total of 77 open baited regions corresponding to 81 gene promoters were linked to 84 open chromatin regions harboring one or more BMD proxy SNPs through 104 distinct chromatin looping interactions (Table 1). When relaxing the $r^2$ threshold for informative SNPs to 0.4, we uncovered additional potentially interesting SNP-promoter interactions for a total of 82 loci (Supplementary Data 4). The vast majority of the implicated genes (78% of the protein coding genes, for which we have reliable RNA-seq expression data) were highly expressed (>50th percentile) in the MSC-derived osteoblasts (Supplementary Data 3).

SNP-promoter interactions (for SNPs not residing in baits, using a threshold of $r^2 > 0.8$ with the sentinel SNP) fell in to three types of observations: (1) to nearest gene only, with respect to location of the sentinel SNP (30%) (*SMAD3*, Fig. 1a), (2) to both nearest and more distant gene(s) (13%) (*CPED1* and *ING3*, Fig. 1b), and (3) only to distant gene(s) (57%) (*EPDR1* and *SFRP4*, Fig. 1c). Coverage tracks and CHiCAGO plots for these loci are shown in Supplementary Figures 3 and 4. Overall, 46% of the GWAS loci interacted with only one baited promoter region, while 54% interacted with more than one.

To evaluate the reliability of gene lists detected in our approach, we went on to carry out pathway analyses on the above genes implicated by ATAC-seq plus Capture C in human MSC-derived osteoblasts (leveraging the liberal $r^2 > 0.4$ threshold in order to achieve more statistical power). Using Ingenuity Pathway Analysis (IPA), we found six enriched canonical pathways ($p < 0.05$, right-tailed Fisher's Exact test with B-H multiple comparison correction), all very relevant to osteoblastic differentiation: role of osteoblasts, osteoclasts and chondrocytes in rheumatoid arthritis, osteoarthritis pathway, role of macrophages, fibroblasts and endothelial cells in rheumatoid arthritis, BMP signaling pathway, TGF-β signaling, and Wnt/β-catenin signaling (Supplementary Figure 5). Among the implicated genes (contacting proxies with $r^2 > 0.8$ with the sentinel GWAS SNP), several have a known role in osteogenesis, such as *SMAD3*, *SMAD9*, *SPP1*, *WLS*, *FRZB*, *NOG*, and *MIR31HG*[46], confirming the validity of our approach, while other target genes are completely novel.

In order to assess cell type specificity of the SNP-promoter interactions uncovered by our approach, we compared SNP-promoter pairs detected from osteoblasts to parallel genome-wide results we had generated from a human liver carcinoma cell line, HepG2, a cell type unrelated to BMD and osteoporosis (Supplementary Tables 3 and 4). In this setting, we only detected interactions for 6 of the 273 BMD loci investigated (Supplementary Data 5); two of them (at *FRZB* and *LEKR1*) were the same interactions detected in osteoblasts; at *LINC00880* the interactions detected in HepG2 were to a subset of the target promoters observed in osteoblasts; one (at *LOC441178*) was a novel interaction, and the last two were at *PLEKHG4*, where a different proxy SNP interacted with one of the targets found in osteoblasts plus a novel target, and at *SLX4IP* where the same proxy SNP interacted with a different target gene. These results highlight the strong cell type specificity of the interactions at the BMD loci investigated.

**Functional studies**. To validate our findings, we performed functional studies of implicated genes at two key BMD loci: *WNT16*, *CPED1*, and *ING3* at the 'WNT16-CPED1' locus, and *EPDR1* and *SFRP4* at the 'STARD3NL' locus. The first is a complex locus, strongly associated with BMD in pediatric cohorts, containing multiple independent signals; in our osteoblast setting, we detected interactions between a region containing two proxies of the BMD DEXA sentinel rs13245690 (rs1861000 and rs3068006, $r^2 = 0.63$ for both) and the *CPED1* and *ING3* promoters. The second locus contains two independent signals, one nearest to *STARD3NL* (rs6959212) and one nearest to *SFRP4* (rs17236800), associated with both DEXA and heel BMD, plus a third independent signal (rs1717731 at *EPDR1*) only detected by heel BMD. Interestingly, we detected interactions between a region containing open proxies of both rs6959212 (rs1524068 and rs940347; $r^2 = 1$) and rs1717731 (rs939666; $r^2 = 0.87$) and the *SFRP4* and *EPDR1* promoters.

We targeted the expression of these genes using siRNA in primary human MSCs derived from four donors (Supplementary Table 1) and then assessed osteoblast differentiation. qPCR analysis showed that each siRNA resulted in significant knock-down of its corresponding target across the donor MSCs, but did not impact the expression of the other gene or genes implicated at the same loci (Supplementary Figure 6 A, B, C, E, F). BMP2 treatment had no effect on *ING3* expression, but reduced *CPED1* and increased *WNT16*. In addition, upon BMP2 treatment, basal expression of *SFRP4* decreased, while *EPDR1* increased, although, considered across donors (which show variability), the results are not statistically significant.

**Table 1 Implicated target genes at 46 BMD GWAS loci in hMSC-derived osteoblasts**

| GWAS Locus | Sentinel SNP | Proxy SNP ($r^2$) | Implicated genes (FPKM) | GWAS Set |
|---|---|---|---|---|
| ANKFN1 | rs72829754 | rs56335503 (0.96), rs8065311 (0.95) | NOG (21.00) | Heel |
| C11orf58 | rs56928337 | rs78152188 (0.99) | TCONS_00019584 (NA), TCONS_00019858 (NA) | Heel |
| CCND1 | rs4980659 | rs6606645 (0.91) | CCND1 (88.53) | DEXA |
| CRADD | rs7969076 | rs7953280 (0.88) | SOCS2 (3.78), SOCS2-AS1 (0.59) | Heel |
| DLX5 | rs1724298 | rs1724298 (1) | SHFM1 (9.38) | Heel |
| DLX6-AS1 | rs17598132 | rs73402679 (0.94) | SHFM1 (9.38) | Heel |
| DNM3 | rs12041600 | rs1992549 (1), rs1992550 (1), rs2586393 (0.99), rs6694387 (0.94) | DNM3OS (8.92), MIR199A2 (NA) | Heel |
| EMP1 | rs76243438 | rs116942315 (1), rs117604090 (1), rs199595633 (1), rs75180224 (1) | EMP1 (30.57) | Heel |
| EN1 | rs62159864 | rs62159869 (0.97) | TCONS_00004419 (NA) | Heel |
| EPDR1 | rs1717731 | rs939666 (0.87) | EPDR1 (25.59), SFRP4 (0.88) | Heel |
| FADS2 | rs174574 | rs3834458 (0.92) | FADS1 (19.09) | Heel |
| FRZB | rs10206992 | rs36090522 (0.99), rs9288087 (0.99) | FRZB (12.92) | Heel |
| GMDS-AS1 | rs4959677 | rs1010762 (0.85) | LINC01600 (NA), TCONS_00011252 (NA) | Heel |
| GNG12-AS1 | rs143243230 | rs143243230 (1) | WLS (12.78) | Heel |
| HOXA-AS3 | rs62454420 | rs62454420 (1) | CBX3 (15.70), SKAP2 (5.32) | Heel |
| HOXA11 | rs17501090 | rs149457254 (0.82) | HIBADH (10.12), TAX1BP1 (9.35) | Heel |
| HOXC6 | rs736825 | rs765634 (0.88) | HOXC-AS1 (0.34), HOXC4 (0.15), HOXC5 (0.36), HOXC9 (8.71), SMUG1 (5.16), TCONS_00020435 (0.02), TCONS_00020436 (NA), TCONS_00020437 (0.01), TCONS_00020438 (NA) | DEXA |
| IGFBP7 | rs11133474 | rs11133472 (0.99), rs11133474 (1) | IGFBP7 (1445.99), IGFBP7-AS1 (0.58) | Heel |
| JAG1 | rs17457340 | rs141094380 (0.95), rs78438678 (0.95) | LOC339593 (NA), TCONS_l2_00016479 (NA), TCONS_l2_00016480 (NA), TCONS_l2_00016851 (0.01) | Heel |
| KCNE2 | rs55787537 | rs55787537 (1) | LINC00649 (0.02), TCONS_00029004 (NA) | Heel |
| KIAA2012 | rs62195575 | rs13000371 (0.82), rs13384015 (0.87) | CDK15 (0.09), KIAA2012 (NA) | Heel |
| KIAA2018 | rs1026364 | rs150722690 (0.90), rs150722690 (0.86), rs16861312 (0.90), rs9288983 (0.90), rs9851731 (0.87) | ATP6V1A (21.00), NAA50 (13.11) | DEXA |
| LEKR1 | rs344081 | rs344088 (0.97) | SSR3 (60.86), TIPARP (11.64), TIPARP-AS1 (0.15) | DEXA |
| LINC00880 | rs56082403 | rs13322435 (0.98), rs56406311 (0.9), rs9817452 (0.9), rs9854955 (0.98) | LEKR1 (0.08), LINC00886 (0.55), SSR3 (60.86), TCONS_00006285 (NA), TCONS_l2_00019596 (0.03), TIPARP (11.64), TIPARP-AS1 (0.15) | Heel |
| LOC101929268 | rs6471752 | rs72639005 (0.84), rs55864946 (0.84), rs72639011 (0.84), rs72639012 (0.84) | LOC101929217 (NA), SNAI2 (39.39), TCONS_00014692 (0.01) | Heel |
| MAPT/WNT3 | rs1864325 | 17:44121917:C:T (0.99), 17:44231617:A:T (0.99), 17:44237372:G:C (0.99), 17:44245359:G:A (0.84), rs62063683 (0.99), rs974291 (0.99), rs974292 (0.99) | KANSL1 (1.89), KANSL1-AS1 (0.71) | DEXA |
| MED13L | rs73200209 | rs17498543 (0.84) | MIR4472-2 (NA) | DEXA |
| MEPE | rs6532023 | rs1471401 (1) | SPP1 (12.86) | Both |
| MLPH | rs58057291 | rs13408361 (0.9), rs6722471 (0.93), rs7569197 (0.9), rs7571898 (0.9), rs7598954 (0.87), rs9287620 (0.87) | COL6A3 (125.76), TCONS_00003518 (NA) | Heel |
| MTAP | rs7035284 | rs7852691 (0.98) | MIR31HG (9.48) | DEXA |
| PHLDB1 | rs10790255 | rs10892247 (0.85) | PHLDB1 (11.37) | Heel |
| PLEKHG4 | rs17680862 | rs8047360 (0.8), rs8050375 (0.8), rs114485334 (0.83) | B3GNT9 (27.79), CES4A (0.22), FBXL8 (2.05), TRADD (11.46) | Heel |
| RERE | rs2252865 | rs2708633 (0.92), rs301789 (0.93), rs301790 (0.93), rs301791 (0.93), rs301792 (0.93) | LOC102724552 (NA), RERE (0.94) | DEXA |
| RNU5F-1 | rs2275707 | rs1694593 (0.96), rs1694594 (0.96) | SLC30A10 (0.00) | Heel |
| SFRP4 | rs17236800 | rs10264106 (1), rs1014939 (0.99), rs2167269 (0.99) | STARD3NL (6.44), TRG-AS1 (NA) | Both |
| SLC1A3 | rs1428968 | rs1428968 (1) | SLC1A3 (0.78) | Heel |
| SLC8A1 | rs10490046 | rs10490046 (1) | SLC8A1 (3.39) | Both |
| SLX4IP | rs6040006 | rs111539884 (0.88), rs74516716 (0.88), rs78532542 (0.88) | LOC339593 (NA), TCONS_l2_00016479 (NA), TCONS_l2_00016480 (NA), TCONS_l2_00016851 (0.01) | Heel |
| SMAD3 | rs1545161 | rs1545161 (1), rs28587205 (0.98), rs72219900 (0.99) | SMAD3 (2.59) | Both |
| SMAD9 | rs556429 | rs17217404 (0.95) | SMAD9 (11.59) | DEXA |
| SMG6 | rs35401268 | rs35401268 (1) | SMG6 (3.19) | Both |
| SPEN | rs6701290 | rs6688051 (0.93) | FLJ37453 (NA), SPEN (1.15), ZBTB17 (6.09) | Heel |
| SPTB | rs1957429 | rs28370916 (0.83) | SPTB (0.02) | DEXA |
| STARD3NL | rs6956946 | rs1524068 (1), rs940347 (1) | EPDR1 (25.59), SFRP4 (0.88) | Both |
| TBPL2 | rs34652660 | rs12431542 (0.90), rs202007979 (0.89), rs4901570 (0.90), rs8009706 (0.90) | KTN1 (10.46), KTN1-AS1 (0.03) | DEXA |
| WLS | rs17482952 | rs17482952 (1), rs79441491 (0.97) | WLS (12.78) | DEXA |

For each locus, proxy SNPs in open chromatin looping to an open promoter are reported, together with their sentinel SNPs and the associated GWAS loci. Looping interactions and open chromatin maps were derived from promoter-focused Capture C and ATAC-seq experiments on hMSC-derived osteoblasts cultures from 3 or more individuals. Average expressions (in FPKM) of the target genes in hMSC-derived osteoblasts from RNA-seq experiments from the same 3 individuals are also reported.

To validate the knock-down of *ING3* and *EPDR1* at the protein level, we performed immunoblot analysis in siRNA transfected cells co-stimulated with BMP2 for 72 h. Each siRNA effectively reduced protein levels of its corresponding target (ING3, ~70% knockdown; EPDR1, ~85% knockdown; Supplementary Figure 7 A-C). Additionally, the influence of *ING3* and *EPDR1* silencing on canonical BMP signaling was tested by evaluating the protein levels of P-SMAD 1/5/8. EPDR1 silencing reduced the levels of P-SMAD 1/5/8, whereas ING3 silencing did not significantly affect SMAD phosphorylation (Supplementary Figure 7 A, D).

Since ALP expression is considered essential for hydroxyapatite deposition and hard-tissue mineralization, histochemical ALP staining was performed using parallel sets of gene-targeted cells. While targeting *WNT16*, *CPED1* and *SFRP4* produced somewhat variable ALP staining across donor lines, staining was strikingly and consistently reduced by *ING3* and *EPDR1* targeting (Fig. 2d, e, j and k). Furthermore, Alizarin red S staining confirmed that calcium phosphate mineral deposition was much reduced in cells with decreased *ING3* and *EPDR1* expression, whereas it could be observed in cells that lacked *CPED1*, *WNT16* and *SFRP4* expression (Fig. 2d, f, j and l). These changes in osteoblast

differentiation while associated with reduced ALP gene expression (Fig. 2c, i), did not impact negatively BMP2 signaling based on *ID1* expression (Fig. 2a, g), and were not associated with concomitant decreases in osteoblast transcriptional regulator *SP7* (Supplementary Figure 6 D, G). Surprisingly, despite the negative impact of *ING3* knockdown on osteoblast differentiation, *ING3* knockdown increased both *ID1* and *RUNX2* expression more than control siRNA. *RUNX2* expression was not significantly affected by *EPDR1* knockdown, although expression was variable among the samples (Fig. 2b, h).

In additional sets of experiments, we tested whether *ING3* and *EPDR1* are specifically involved in osteoblast differentiation or simply represent essential genes for general cell metabolism, cell division or DNA replication. We first investigated chromatin interactions at these two loci in MSC-derived adipocytes. Interestingly, in this setting we observed the same interactions seen in osteoblasts to the *EPDR1* and *ING3* promoters, while we did not detect interactions to the *SMAD3* promoter (Supplementary Data 6). We therefore assessed the impact of the *ING3* and *EPDR1* siRNA knockdowns in MSC during adipogenic differentiation. *ING3* and *EPDR1* silenced cells were cultured under

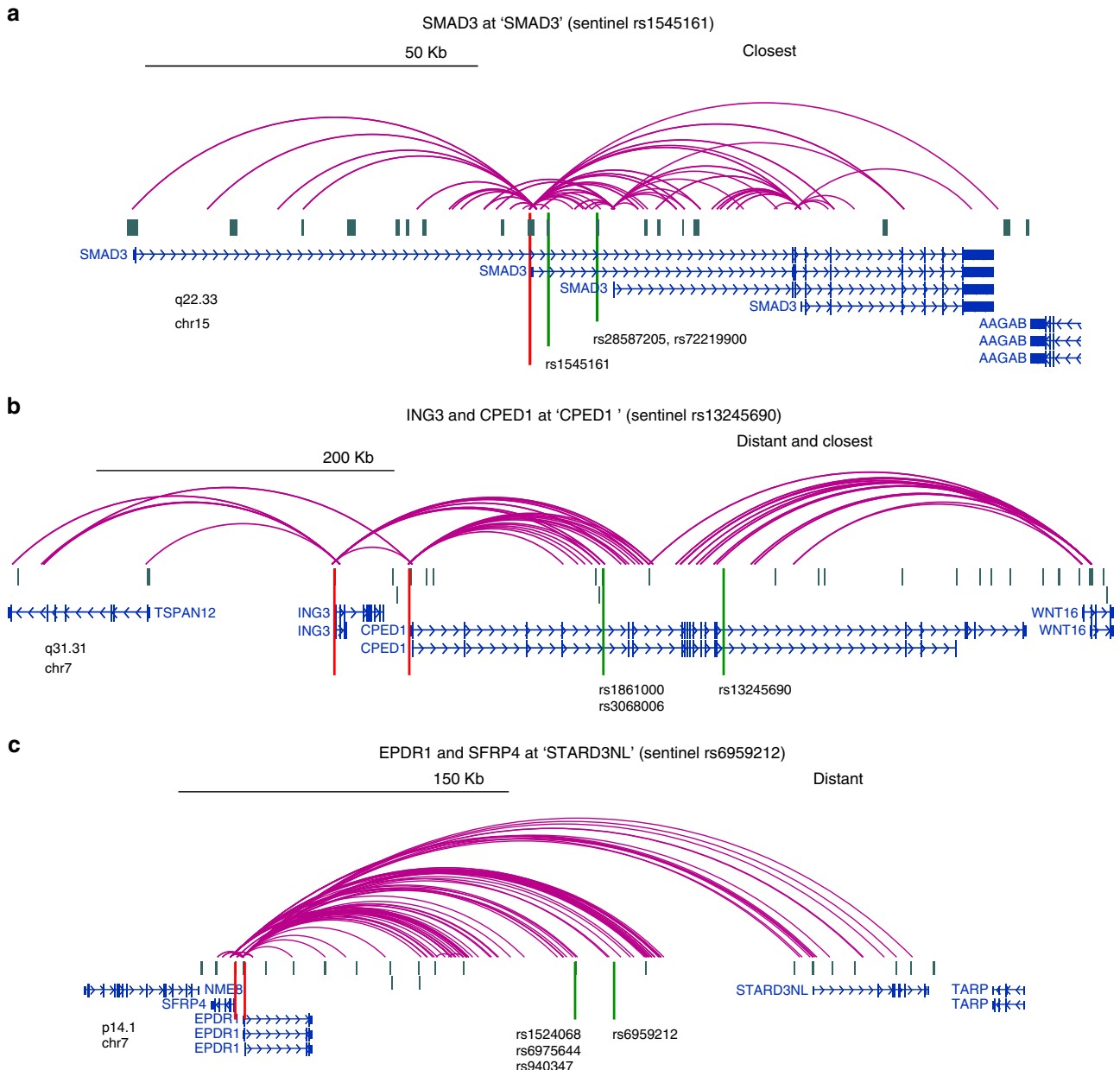

**Fig. 1** Examples of interactions detected by Capture C. **a** to nearest gene only. **b** to both nearest and more distant gene(s). **c** only to distant gene(s). Contacts were visualized using the WashU EpiGenome Browser. Red lines indicate genes of interest, while green lines represent the sentinel and key proxy SNPs

adipogenic conditions for 3 weeks and accumulation of lipid droplets examined by oil red o staining. Remarkably, silencing of each of these genes enhanced adipogenic differentiation (*ING3* knockdown increased adipogenesis by ~8 fold, and *EPDR1* knockdown by ~3.5 fold; Fig. 3a, b and Supplementary Figure 4 H, I) suggesting a reciprocal relationship of these genes with regards to adipogenesis and osteoblastogenesis. Although not shown in the current study, foci resembling lipid droplets could be detected as early as 1 week in both *ING3* or *EPDR1* targeted cells, whereas such foci could only be detected after 2 weeks of adipogenic induction in control cells. Supporting this observation, gene expression analysis revealed that the expression of the *C/EBP alpha* gene (an adipogenic transcription factor) was significantly increased in *ING3* and *EPDR1* silenced cells (Fig. 3c). The levels of another adipogenic transcription factor PPAR gamma was also evaluated, which was consistently higher but did

not quite reach statistical significance in *ING3* and *EPDR1* targeted cells (Fig. 3d).

In summary, knock-down of two novel genes (*ING3* and *EPDR1*)—not previously associated with BMD but implicated by our combined ATAC-seq and Capture C approach —revealed strong reciprocal effects on osteoblast differentiation (decreased ALP expression and absence of calcium phosphate mineral deposition) and adipogenic differentiation (enhanced accumulation of lipid droplets and increased expression of C/EBP alpha).

## Discussion

In this study, we applied genome-wide promoter-focused Capture C to primary human MSC-derived osteoblasts, a highly relevant cellular model for BMD and osteoporosis. By combining Capture C technique with ATAC-seq, we were able to map 95 variants (corresponding to 46 BMD GWAS loci) to 81 gene promoters.

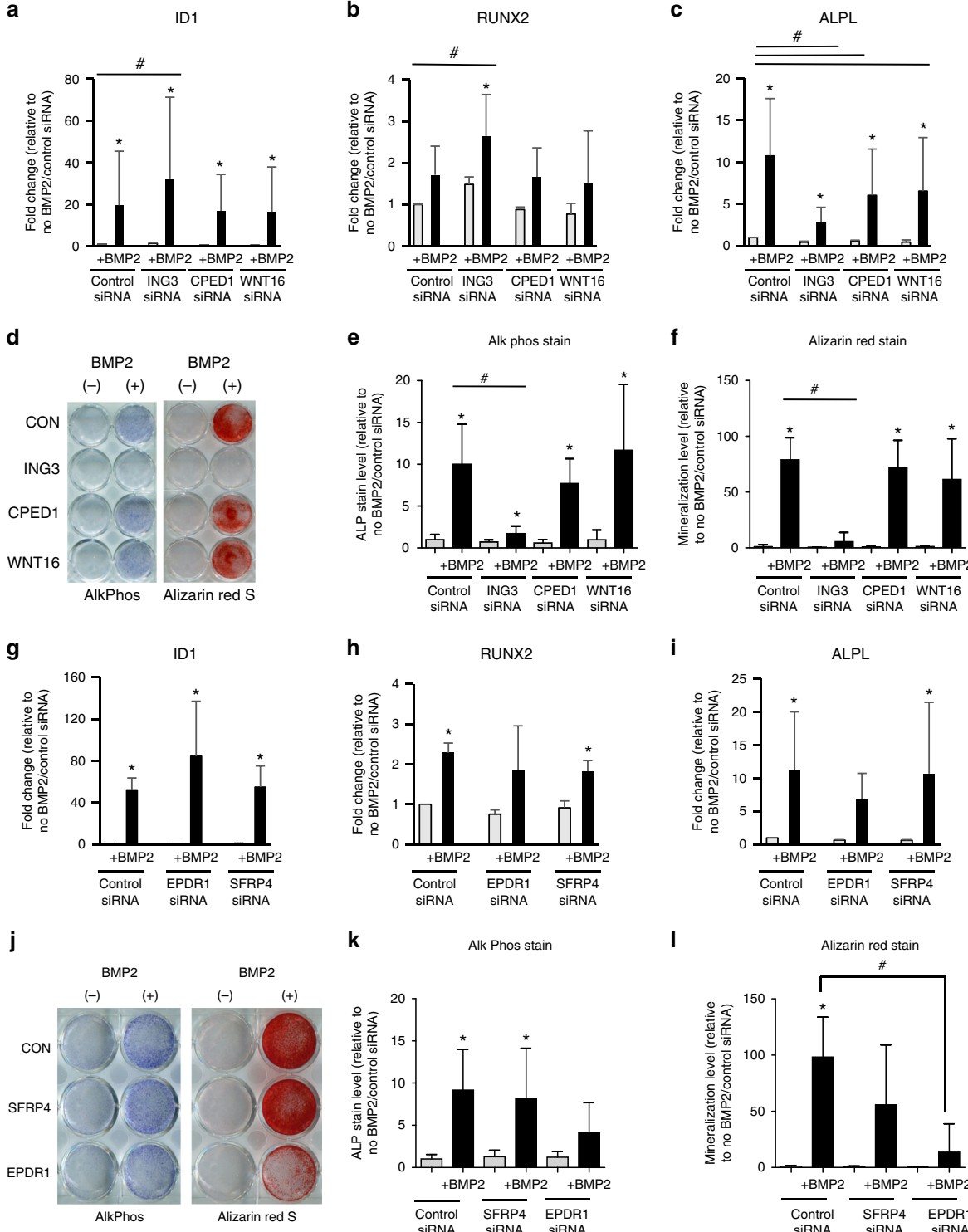

**Fig. 2** Knockdowns of *ING3* and *EPDR1* impair osteoblast differentiation. *ING3* knockdown at the GWAS implicated *CPED1* locus results in a complete disruption of alkaline phosphatase induction and Alizarin red S staining, but there is no effect of *CPED1* and *WNT16* knockdown. Similarly, *EPDR1* disruption at *STARD3NL* results in a reduction in alkaline phosphatase expression and activity and alizarin red S staining, but there is no effect of *SFRP4* knockdown. **a–c**, **g–i** Quantitative gene expression. Gray columns = No BMP treatment; Black columns = BMP treatment. Columns = mean. Error bars = Standard deviation. $n = 4$ (for *ING3, CPED1,* and *WNT16* datasets) and $n = 3$ (for *SFRP4* and *EPDR1* datasets) unique donor lines. *$p < 0.05$ comparing No treatment to BMP treatment for each siRNA. #$p < 0.05$ comparing control siRNA to siRNA for gene of interest (two-way homoscedastic Student's *t*-tests).
**d**, **j** Representative AlkPhos (purple) and Alizarin (red) stained plates, and **e**, **f**, **k**, and **l** quantitative image analysis of staining results repeated with four different independent hMSC donor cell lines. Source data are provided as a Source Data file

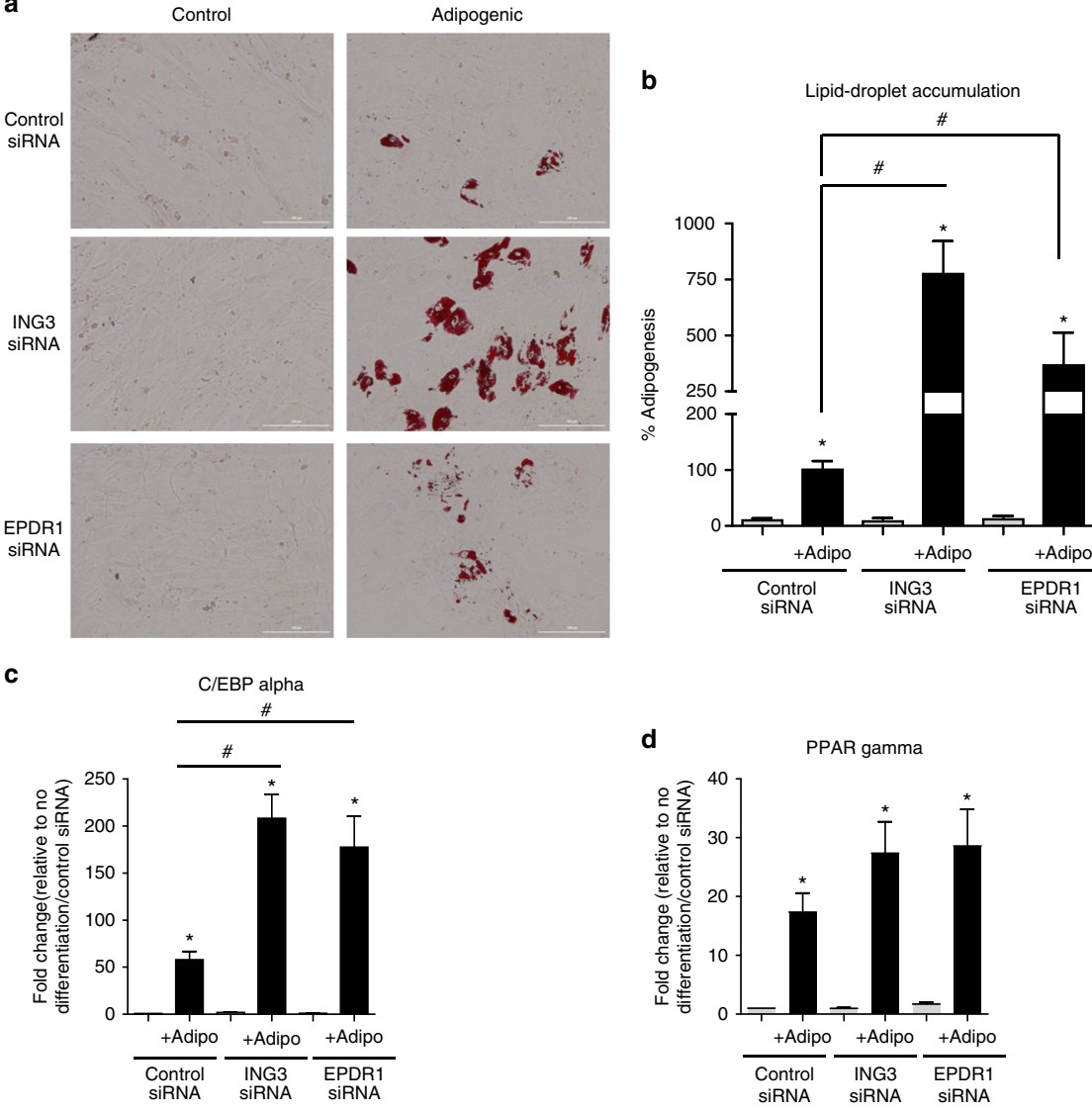

**Fig. 3** *ING3* and *EPDR1* knockdown increases adipogenic differentiation of hMSCs. **a** ING3 and EPDR1 siRNA transfected cells were stimulated with adipogenic differentiation medium for 21 days and stained with Oil Red O for visualization of lipid droplets. Representative results of two hMSC donor lines are shown (scale bar = 200 μm). **b** Relative number of lipid droplets (>25 nm diameter) from 100 different microscopic fields were enumerated from three different experiments (two donor lines) and presented as % adipogenesis compared to control samples. **c, d** Quantitative gene expression. Gray columns = No differentiation; Black columns = Adipogenic differentiation for 1 week (Adipo). Columns = mean. Error bars = standard deviation. $n = 3$ unique donor lines. *$p < 0.05$ comparing no treatment to adipogenic differentiation for each siRNA. #$p < 0.05$ comparing control siRNA to siRNA for gene of interest (two-way homoscedastic Student's $t$-tests). Source data are provided as a Source Data file

This allowed us to determine causal variants for BMD and discover a list of novel genes that physically interacts with causal variants at the 3D genome level. Two novel genes, *ING3* and *EPDR1*, were selected and functionally verified with strong effects on osteoblastic and adipogenic differentiation.

Recently, many chromatin conformation capture (3C) based methods have been developed with the goal of mapping GWAS variants to their target genes[40–42,47–49]. While Hi-C was developed to study the high order genomic organization of human chromatin domains, and not the precise looping interactions between GWAS-implicated variants and their target genes (and requires a high amount of sequencing), the main limitations of the other currently available approaches are the low resolution (dictated by the use of the 6-cutter HindIII) of Capture Hi-C[41], and hypothesis-driven nature of the Hi-ChIP technique[49], which is not promoter-centric and uses antibodies to capture genomic

regions characterized by certain histone marks (such as H3K27ac). To overcome these limitations, and given the paucity of bone-related cell types represented in the public domain (ENCODE, GTEx[37] etc.), we employed a very nimble genome-scale, promoter-focused high-resolution Capture C technique that does not require the large sample sizes typically required for eQTL analysis, providing significant results with only a few biological replicates. This technique allows for a non-hypothesis driven gene discovery effort, since it presumes nothing about the region in which the SNP resides.

Typical of Capture C, our study produced around 5% of 'usable' data (unique di-tags) from the original sequencing reads, which compares relatively unfavorably with other methods such as promoter capture Hi-C which typically yields approximately 25% unique di-tags. However, given the fact that the approach we employed yields much higher resolution, we believe the relative

high sequencing cost returned significantly more information. Indeed, while other comparable methods can also be scaled to target promoters of non-coding genes or alternative promoters, combined with the high resolution we achieve, this approach returns extremely informative data.

Despite the fact that we selected primary human MSC-derived osteoblasts as cellular model in this study, we recognized that other cell types, most particularly osteoclasts, are also involved in BMD determination, but we have shed light on how these loci operate in this key single cell type, most responsible for building peak bone density.

We were able to determine complex and intricate contacts for ~17% of the BMD loci investigated (~30% when relaxing the $r^2$ threshold for informative proxies to 0.4), frequently detecting two or more promoter contacts at these signals. About 70% of the ATAC-seq implicated BMD proxy SNPs not residing in a promoter interacted with distal genes, either exclusively or in conjunction with an interaction with the nearest gene. Implicated genes revealed enrichment for functionally relevant networks for bone biology. We went on to demonstrate with siRNA mediated knockdown experiments that *ING3* and *EPDR1*, implicated genes at the 'WNT16-CPED1' and 'STARD3NL' loci, respectively, play a role in osteoblast differentiation. Most intriguingly, neither gene is widely known to the bone community and thus reveals novel biology. Interestingly, both *EPDR1* and *ING3* knock-outs had opposite effects on hMSC differentiation to osteoblasts (impaired) or adipocytes (enhanced). These results imply that both *ING3* and *EPDR1* are causal genes that modulate the fate of hMSC differentiation, rather than just having an indirect effect on cell metabolism and proliferation, and suggest an important role in bone biology and have implications for the development of diagnostic and therapeutic tools for osteoporosis.

Although our studies have not probed the mechanistic basis for the regulation of osteoblastogenesis by *ING3* and *EPDR1*, prior studies in other organ systems may provide some clues regarding potential function. Two mechanisms can be proposed by which *ING3* affects osteoblast differentiation. First, since the ING3 protein is part of the NuA4 histone acetyltransferase (HAT) complex that recognizes trimethylated forms of lysine 4 of histone H3 (H3K4me3)[50], silencing of *ING3* might affect key osteoblastic genes during human osteoblast differentiation. Second, given that ING3 levels are down-regulated in head and neck carcinoma, melanoma, ameloblastoma, hepatocellular carcinoma, and colorectal cancers[50], and ING3 containing transcription complexes interact with p53 transactivated promoters including promoters of p21/WAF1 and BAX to affect cell cycle progression[51], the absence of ING3 may allow cells to avoid differentiation by continuously re-entering the cell cycle; however, our studies are carried out in serum-free media and thus, we did not note any changes in cell proliferation, plus we showed that *ING3* knock-out actually promotes adipogenic differentiation. EPDR1 is a type II transmembrane protein that is similar to two families of cell adhesion molecules, the protocadherins and ependymins that can affect the extracellular milieu (ECM). Calcium-induced conformational change of EPDR1 molecules are known to be important for EPDR1 interaction with the components of the extracellular matrix and this can affect cell adhesion and migration[52]. Notably, extracellular matrix components play an important role in osteoblast differentiation, and thus, alterations in interactions of EPDR1 with the ECM could have important implications for this biological process. Intriguingly, an *Epdr1* KO mouse exists in the Mouse Genome Database (MGD) Project and has a short tibia phenotype.

We cannot exclude the role of other genes at these loci. At 'WNT16-CPED1', the culprit gene is usually believed to be *WNT16*, since the WNT pathway plays a fundamental role in bone development and Wnt16 has been shown to regulate cortical bone mass and bone strength in mice[53]. Indeed, two SNPs in LD with the *WNT16* GWAS sentinel rs3801387 (rs142005327, $r^2 = 0.94$; rs2908004, $r^2 = 0.45$) reside in the *WNT16* promoter region and may well regulate WNT16 expression; however, our Capture C data reveal no looping between this promoter and putative enhancer open chromatin regions in our cell setting.

At the 'STARD3NL' locus, a compelling candidate effector gene is *SFRP4*, encoding secreted frizzled-related protein 4, a soluble Wnt inhibitor recently implicated in bone remodeling both in mouse and humans[54,55]. Nonetheless, and most interestingly, our approach was able to uncover the role of two less obvious, novel genes in osteoblastogenesis and bone mineralization.

In conclusion, we observed consistent contacts at multiple BMD GWAS loci with a high-resolution promoter interactome applied to a single difficult to obtain, disease-relevant cell type i.e. human primary MSC-derived osteoblasts. This ATAC-seq plus promoter-focused Capture C approach holds promise for future efforts to implicate effector genes at GWAS loci for other common genetic disorders.

## Methods

**Loci analyzed**. We leveraged 63 loci from the latest GWAS of adult BMD and fracture[28], plus pediatric and adult loci from studies from our and other investigators' groups[24,25,28,30,38,56,57], for a total of 110 independent sentinels for 107 DEXA BMD loci. We also investigated 307 independent signals from 203 loci derived from a large GWAS of heel ultrasound BMD[39] (Supplementary Data 1). We did not utilize the SNP list from the latest ultrasound BMD metanalysis[23] given this dataset became available after the work had been commenced. To obtain proxy SNPs, we used rAggr (http://raggr.usc.edu) with an $r^2 < 0.8$ threshold and the all European (CEU+FIN+GBR+IBS+TSI) population.

**Human Mesenchymal Stem Cells**. Primary bone-marrow derived human MSCs isolated from healthy donors (age range: 22–29 years) were characterized for cell surface expression (CD166+CD90+CD105+/CD36−CD34−CD10−CD11b−CD45−) and tri-lineage differentiation (osteoblastic, adipogenic and chondrogenic) potential at the Institute of Regenerative Medicine, Texas A&M University. Expansion and maintenance of the cells were carried out using alpha-MEM supplemented with 16.5% FBS in standard culture conditions by plating cells at a density of 3000 cells/cm².

For osteoblastic differentiation, 15,000 cells/cm² from maintenance cultures were plated in alpha-MEM consisting of 16.5% FBS, 25 μg/ml Ascorbic acid-2-phosphate, 5 mM beta-glycerophosphate and 1% insulin-transferrin-selenous acid (osteogenic media) and stimulated the next day with recombinant human BMP2 (300 ng/ml) (R&D Systems, MN) in serum-free osteogenic media. Cells were harvested at 72 h following BMP2 treatment for sequencing library preparations because our previous work has shown that this time point reflects a stage when the cells are fully osteoblast committed but have not begun to mineralize. It was important to harvest non-mineralizing cultures, given that during that terminal state cells will begin to undergo apoptosis[43,58]. Cells were assessed for differentiation in parallel plates by harvesting RNA and assessing the expression of RUNX2, SP7, and alkaline phosphatase, and target genes. Additionally, a third parallel plate of cells was assessed for alkaline phosphatase activity. For quantitative RT-PCR analysis, 300 ng of purified total RNA was reverse transcribed using High Capacity cDNA Reverse Transcription Kit (Applied Biosystems) in a 20 μl reaction. One microliter of the resulting cDNA was amplified using Power SYBR® Green PCR Master Mix and gene-specific primers in a 7500 Fast Real-Time PCR System (Applied Biosystems) following manufacturer's recommendations. For assessing mineralization, cells were analyzed 8–10 days after BMP stimulation by staining with Alizarin red S. All values are reported as mean ± standard deviation with statistical significance determined via 2-way homoscedastic Student's $t$-tests (*$P ≤ 0.05$, #$P ≤ 0.10$, N.S. = "not significant").

Adipogenic differentiation of the hMSCs was carried out 2 days after seeding using standard procedure for human cells. Briefly, cell monolayers were washed twice with phosphate buffered saline and supplemented with alpha-MEM consisting of 8.25% FBS, 0.5 μM water-soluble Dexamethasone, 0.5 μM isobutylmethylxanthine, and 50 μM Indomethacin. Alpha-MEM consisting of 8.25% FBS with or without supplementation were exchanged every 2–3 days until harvested (2 weeks induction for capture C library preparation, 1 week induction for gene expression analysis, and 3 weeks for oil red o staining).

**Quantification of Alk Phos and Alizarin Red Staining**. After histochemical staining for the alkaline phosphatase activity and calcium deposition into the extracellular matrix by Alizarin Red staining, multi-well plates were allowed to air-dry and each well was scanned using high-resolution color brightfield objective

(1.25X) of the Lionheart FX automated microscope (BioTek). For each scanned well, Image analysis was performed using Image J software according to the guidelines provided by the National Institute of Health. Briefly, images were converted to 8-bit grayscale and integrated pixel values were measured within a region of interest (ROI) covering entire culture area using same detection threshold for each multi-well plate. Data was combined from at least 3 different donor lines and represented as average fold change compared to control siRNA transfected cells without BMP2 stimulation.

**Immunoblotting**. Immunoblotting was performed using standard procedures in siRNA transfected cells[59] with minor modifications. Briefly, hMSC monolayers on 12-well plates were washed three times with ice-cold PBS, and 100 µl of lysis buffer composed of 50 mM Tris-Cl, 150 mM NaCl, 0.1% SDS, 0.1% Igepal CA 630, 0.5% sodium deoxycholate, and protease inhibitor cocktail (Roche) was added. Cell lysates were collected, vortexed vigorously, and clarified by centrifugation. The protein concentrations in the supernatant were determined using BCA protein assay (Pierce). Ten micrograms of each lysate were loaded into SDS-polyacrylamide gels and electro transferred onto polyvinyl difluoride membranes. Membranes were blocked for 1 h in 2.5 % non-fat skim milk in T-TBS (Tris-buffered saline containing 0.01% Tween-20), then incubated overnight at 4 ˚C with primary antibodies at the dilutions suggested by the manufacturers (see below for antibody information). Membranes were washed three times with T-TBS, then incubated with horse-radish-peroxidase conjugated secondary antibodies for 1 h at room temperature. Finally, the blots were incubated for 5 min in Supersignal™ West Femto Chemiluminescent Substrate (Fisher Scientific) and data were captured on Bio-Rad Chemi Doc system using appropriate settings for each antibody. Relative band intensities from each blot were calculated using Image Lab software v5.2.1 (Bio-Rad) and data from 4 different donor lines were combined for statistical analysis. The following primary and secondary antibodies were used: ING3 clone 2A2 (Millipore, MABC1185, 1:3000); EPDR1 (Abcam, ab197932, 1:1000), phospho-SMAD 1/5/9 (Cell Signaling Technology, 13820S, 1:1000), SMAD5 (Cell Signaling Technology, 9517S, 1:1000), GAPDH (Cell Signaling Technology, 5174S, 1:30000), Anti-rabbit IgG, HRP-linked antibody (Cell Signaling Technology, 7074S, 1:5000), Anti-mouse IgG, HRP-linked antibody (Cell Signaling Technology, 7076S, 1:5000).

**ATAC-seq library generation and peak calls**. Tn5 Transposase transposition (Illumina Cat #FC-121–1030, Nextera) and purification of the Tn5 Transposase derived DNA from 100,000 human MSC-derived osteoblasts or adipocytes was performed at the University of Michigan. The samples were then shipped to the Center of Spatial and Functional Genomics at CHOP where the ATAC-seq process was completed. Live cells were harvested via trypsinization, followed by a series of wash steps. 100,000 cells from each sample were pelleted at $550 \times g$ for 5 min at 4 °C. The cell pellet was then resuspended in 50 µl cold lysis buffer (10 mM Tris-HCl, pH 7.4, 10 mM NaCl, 3 mM MgCl2, 0.1% IGEPAL CA-630) and centifuged immediately at 550 ×g for 10 min at 4 °C. The nuclei were resuspended in the transposition reaction mix (2x TD Buffer (Illumina Cat #FC-121–1030, Nextera), 2.5 µl Tn5 Transposase (Illumina Cat #FC-121–1030, Nextera) and Nuclease Free H$_2$O) on ice and then incubated for 45 min at 37°. The transposed DNA was then purified using the MinElute Kit (Qiagen), eluted with 10.5 µl elution buffer (EB), frozen and sent to the Center for Spatial and Functional Genomics at CHOP. The transposed DNA was PCR amplified using Nextera primers for 12 cycles to generate each library. The PCR reaction was subsequently cleaned up using AMPureXP beads (Agencourt) and libraries were paired-end sequenced on an Illumina HiSeq 4000 (100 bp read length) and the Illumina NovaSeq platform. Open chromatin regions were called using the ENCODE ATAC-seq pipeline (https://www.encodeproject.org/atac-seq/), selecting the resulting IDR conservative peaks (with all coordinates referring to hg19). We define a genomic region open if it has 1 bp overlap with an ATAC-seq peak.

**Cell fixation for chromatin capture**. The protocol used for cell fixation was similar to previous methods[41]. Cells were fixed after 3 days of BMP2 stimulation or after 2 weeks of adipogenic induction. Cells were collected and single-cell suspension were made with aliquots of 10$^7$ cells in 10 ml media (e.g. RPMI + 10%FCS). 540 µl 37% formaldehyde was added and incubated for 10 min at RT on a platform rocker. The reaction was quenched by adding 1.5 ml 1 M cold glycine (4 °C) for a total volume of 12 ml. Fixed cells were centrifuged at 1000 rpm for 5 min at 4 °C and supernatant removed. The cell pellets were washed in 10 ml cold PBS (4 °C) followed by centrifugation as above. Supernatant was removed and cell pellets were resuspended in 5 ml of cold lysis buffer (10 mM Tris pH8, 10 mM NaCl, 0.2% NP-40 (Igepal) with protease inhibitor cocktails). Resuspended cells were incubated for 20 min on ice, centrifuged as above, and the lysis buffer removed. Finally, cell pellets were resuspended in 1 ml fresh lysis buffer, transferred to 1.5 ml Eppendorf tubes and snap frozen (ethanol/dry ice or liquid nitrogen). Cells were stored at −80 °C until they were thawed for 3C library generation.

**3C library generation**. 3C libraries were generated from fixed MSC-derived osteoblasts or adipocytes shipped from Michigan to CHOP. For each library, 10 million fixed cells were thawed on ice, pelleted by centrifugation and the lysis buffer

removed. The cell pellet was resuspended in 1mL dH2O supplemented with 5uL protease inhibitor cocktail (200X) and incubated on ice for 10 min, followed by centrifugation and removal of supernatant. The pellet was then resuspended in dH2O for a total volume of 650 uL. Control 1 (50uL) was removed at this point and remaining samples were divided into 6 tubes. NEBuffer DpnII (1X), dH2O, and 20% SDS were added and samples were incubated at 1000 rpm for 1 h at 37 °C in a MultiTherm (Sigma-Aldrich). Following the addition of Triton X-100 (concentration, 20%), samples were incubated an additional hour. Next, 10 µL of DpnII (50 U/µL) (NEB) was added and incubated for 4 h at 37 °C. An additional 10 µL DpnII was added and digestion was left overnight. The next day, another 10 µL of DpnII was added and incubated for a further 3 h. 100uL of each digestion reaction was removed to generate control 2. Finally, samples were incubated at 1000 rpm for 20 min at 65 °C to inactivate the DpnII and placed on ice for 20 additional minutes.

Next, the digested samples were ligated with 8uL T4 DNA Ligase (HC ThermoFisher, 30 U/µL) and 1X ligase buffer at 1,000 rpm overnight at 16 °C in the MultiTherm. The next day, an additional 2 µL T4 DNA ligase was spiked in to each sample and incubated for 3 more hours. The ligated samples were then de-crosslinked overnight at 65 °C with Proteinase K (Invitrogen) and the following morning incubated for 30 min at 37 °C with RNase A (Millipore). Phenol-chloroform extraction was then performed, followed by ethanol precipitation overnight at −20 °C and a 70% ethanol wash. Samples were resuspended in 300uL dH2O and stored at −20 °C until Capture C. Digestion efficiencies of 3C libraries were assessed using control 1 (undigested) and control 2 (digested) by gel electrophoresis on a 0.9% agarose gel and quantitative PCR (SYBR green, Thermo Fisher).

**Capture C**. Custom capture baits were designed using an Agilent SureSelect library design targeting both ends of DpnII restriction fragments encompassing promoters (including alternative promoters) of all human coding genes, noncoding RNA, antisense RNA, snRNA, miRNA, snoRNA, and lincRNA transcripts, totaling 36,691 RNA baited fragments. The library was designed using scripts generously provided by Dr. Hughes (Oxford, UK), utilizing the RefSeq, lincRNA Transcripts and sno/miRNA tracks in the hg19 assembly downloaded from the UCSC Table Browser on 9/16/2015. The capture library design covered 95% of all coding RNA promoters and 88% of RNA types described above. The missing 5% of coding genes that could not  be designed were either duplicated genes or contained highly repetitive DNA in their promoter regions.

The isolated DNA from the 3C libraries was quantified using a Qubit fluorometer (Life Technologies), and 10 µg of each library was sheared in dH$_2$O using a QSonica Q800R to an average DNA fragment size of 350 bp. QSonica settings used were 60% amplitude, 30 s on, 30 s off, 2-min intervals, for a total of five intervals at 4 °C. After shearing, DNA was purified using AMPureXP beads (Agencourt). Sample concentration was checked via Qubit fluorometer and DNA size was assessed on a Bioanalyzer 2100 using a 1000 DNA Chip. Agilent SureSelect XT Library Prep Kit (Agilent) was used to repair DNA ends and for adaptor ligation following the standard protocol. Excess adaptors were removed using AMPureXP beads. Size and concentration were checked again before hybridization. 1 µg of adaptor ligated library was used as input for the SureSelect XT capture kit using their standard protocol and our custom-designed Capture C library. The quantity and quality of the captured library were assessed by Qubit fluorometer and Bioanalyser using a high sensitivity DNA Chip. Each SureSelect XT library was initially sequenced on 1 lane HiSeq 4000 paired-end sequencing (100 bp read length) for QC. All six Capture C promoter interactome libraries were then sequenced three at a time on an S2 flow cells on an Illumina NovaSeq, generating ~1.6 billion paired-end reads per sample (50 bp read length).

**Analysis of Capture C data**. Quality control of the raw fastq files was performed with FastQC. Paired-end reads were pre-processed with the HiCUP pipeline[60], with bowtie2 as aligner and hg19 as reference genome. Significant promoter interactions at 1-DpnII fragment resolution were called using CHiCAGO[44] with default parameters except for binsize which was set to 2500. Signifcant interactions at 4-DpnII fragment resolution were also called with CHiCAGO using artificial . baitmap and .rmap files where DpnII fragments were grouped into four consecutively and using default parameters except for removeAdjacent which was set to False. We define PIR a promoter-interacting region, irrespective of whether it is a baited region or not. The CHiCAGO function peakEnrichment4Features() was used to assess enrichment of genomic features in promoter interacting regions at both 1-fragment and 4-fragment resolution. Finally, we made use of the Washington Epigenome Browser (https://epigenomegateway.wustl.edu) to visualize the detected interactions within the context of other relevant functional genomics annotations.

**RNA-seq**. Total RNA was isolated from differentiating osteoblasts using TRIzol reagent (Invitrogen) following manufacturer instructions, and then depleted of rRNA utilizing the Ribo-Zero rRNA Removal Kit (Illumina). RNA-seq libraries were prepared using the NEBNext Ultra II Directional RNA Library Prep Kit for Illumina (NEB) following standard protocols. Libraries were sequenced on one S2 flow cell on an Illumina NovaSeq 6000, generating ~200 million paired-end 50 bp reads per sample. RNA-seq data were aligned to the hg19 genome with STAR v.

2.5.2b[61] and pre-processed with PORT (https://github.com/itmat/Normalization) using the GENCODE Release 19 (GRCh37.p13) annotation plus annotation for lincRNAs and sno/miRNAs from the UCSC Table Browser (downloaded 7/7/2016). Normalized PORT counts for the uniquely mapped read pairs to the sense strand were additionally normalized by gene size and the resulting values were used in the computation of gene expression percentiles.

**Functional characterization of candidate genes**. We investigated five key genes, residing across two loci—*EPDR1* and *SFRP4* at the '*STARD3NL*' locus and *WNT16*, *CPED1* and *ING3* at the '*WNT16-CPED1*' locus. Experimental knock down of these genes was achieved using DharmaFECT 1 transfection reagent (Dharmacon Inc., Lafayette, CO) using a set of 4 ON-TARGETplus siRNA (target sequences in Supplementary Table 1) in three temporally separated independent MSC-derived osteoblast samples and then assessed for metabolic and osteoblastic activity. Following siRNA transfection, cells were allowed to recover for 2 days and stimulated with BMP2 for additional 3 days in serum-free osteogenic media, as previously described, after which the influence of knockdown on gene expression (qPCR) and early osteoblast differentiation (ALP) were evaluated as described.

Adipogenic differentiation of the *ING3* and *EPDR1* siRNA transfected cells were performed after 2 days of siRNA transfection. The cell monolayers were washed twice with phosphate buffered saline and supplemented with adipogenic induction media until harvested (for gene expression analysis-cells were harvested after 1 week, and for Oil red o staining-cells were harvested at 3 weeks). Quantification of the relative number of lipid droplet for each condition was performed with a Lionheart FX automated microscope. Briefly, 100 color brightfield images from each oil red o stained well were captured using 10X objective and cell count algorithm of the Gene 5 software v3.04 (BioTek) was applied in the green detection channel for objects ranging from 35–100 μM using appropriate detection threshold. Data from two independent donor line (three separate experiments) were combined and represented as 100% adipogenesis for control siRNA transfected cells cultured under adipogenic condition.

## Data availability

Data that support the findings of this study have been deposited in ArrayExpress with the following accession numbers: Capture C: "E-MTAB-6862 [https://www.ebi.ac.uk/arrayexpress/experiments/E-MTAB-6862/]" (MSC-derived osteoblasts) and "E-MTAB-7144 [https://www.ebi.ac.uk/arrayexpress/experiments/E-MTAB-7144/]" (HepG2); ATAC-seq: "E-MTAB-6834 [https://www.ebi.ac.uk/arrayexpress/experiments/E-MTAB-6834/]" (MSC-derived osteoblasts) and "E-MTAB-7543 [https://www.ebi.ac.uk/arrayexpress/experiments/E-MTAB-7543/]" (HepG2); RNA-seq: "E-MTAB-6835 [https://www.ebi.ac.uk/arrayexpress/experiments/E-MTAB-6835/]" (MSC-derived osteoblasts). All other data are contained within the article and its supplementary information. All the code used in this study is publicly available from the cited references. The source data underlying Figs. 2 and 3 and Supplementary Figures 6 and 7 are provided as a Source Data file.

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

## Acknowledgements

This research was funded by the Children's Hospital of Philadelphia and by NIH grant R01 HG010067. Some of the materials employed in this work were provided by the Texas A&M Health Science Center College of Medicine Institute for Regenerative Medicine at Scott & White through a grant from ORIP of the NIH, Grant #P40OD011050. Dr. Grant is funded by the Daniel B. Burke Endowed Chair for Diabetes Research. We thank Dr. Hughes for providing the library design scripts and for useful discussions.

## Author contributions

A.C., M.E.J., E.M., A.D.W. and S.F.A.G. conceived and designed the ATAC-seq plus Capture C variant to gene mapping approach, S.L., M.E.L., K.M.H. and J.A.P. generated the Capture C and ATAC-seq datasets, A.C., E.M. and C.S. performed the bioinformatics analyses, Y.W. and K.D.H. conceived and designed the functional studies, YW preformed the functional studies, A.C., Y.W., M.E.J., E.M., K.D.H., A.D.W. and S.F.A.G. wrote the paper. All authors discussed the results and implications and commented on the manuscript at all stages.

## Additional information

**Competing interests:** The authors declare no competing interests.

