## [Peer Review File · Nature Communications]

Reviewer #1 (Remarks to the Author):

The paper has used promoter Capture-C, ATAC-seq and RNA-seq on a novel, primary cell type (osteoblasts) for a BMD focused analysis. They demonstrate very nicely the utility of this study design, and show how it can be used to direct GWAS findings to novel genes, not necessary the ones implicated by distance/previous knowledge. As such I feel it could be an important and relevant piece of work, not only to this field but to a wider 'GWAS' community.

The work to "validate" the implicated genes is good, using siRNA to knock out novel genes and show they have a potential affect in traits relevant to BMD.

One of the claims in the paper is to have developed a new methodology. I feel the methodology is Capture C, enriched with Agilent baits designed for promoters - so Capture C. I'm not sure it deserves a new acronym.

Some of the results section feel more like introduction/methods, for example lines 105 to 110.

The major issue is with the use of GWAS data. GWAS SNPs are called 'independent' if the $r^2 < 0.2$ and proxy if the $r^2 > 0.4$. There are now robust, statistical methods to define independent affects in a GWAS region, conditioning on the lead variant. More troubling is 'proxy SNPs' of $r^2 > 0.4$. This type of functional analysis of GWAS loci should provide robust evidence of causality on robust GWAS hits. I'm not sure any publications use $r^2 > 0.4$ to define proxy SNPs for GWAS? More usually it is $r^2 > 0.8$, and preferably 0.9. Using $r^2 > 0.4$, I feel, undermines the original GWAS results, such that the functional data is left annotating SNPs that are 'not really associated with BMD'. The 14,007 BMD SNPs, that are used to annotate ones linked to genes, are likely to be a massive overestimate, and the analysis should be performed on a more realistic ($r^2 > 0.8$) set of SNPs, to provide SNPs that genuinely have a chance of being causal, and are linked to genes.

The authors compare their method to other published methods (I think its wrong to imply HiChIP is 'biased' - it asks a different question to this study design, for example about how active enhancers interact with promoters. Maybe hypothesis driven?).

I feel the inefficiencies in this methodology are being compensated for by a large sequencing effort. There is a wealth of techniques available for this work now, and I think its important both the strengths and limitations are clearer for people to make an informed decision of which study design to select. For example, typical of Capture C this study produces around 5% of 'usable' data (Unique di-tags) from the original sequencing reads, which compares unfavourable with other methods (for example Javierre et al Cell 2016 - promoter capture HiC had around 25% unique di-tags). For this type of interaction data typically 10,000 di-tags /bait are optimal (Javierre had ~20,000). Here it is around 3,000/bait (hence the requirement for pooling to call longer interactions). So even though they employed the massive throughput (and expensive) NovaSeq, the study still produced a limited number of unique di-tags per fragment. There may well be advantages of this methodology - lower cell number?, quicker?, but these need to be explained in the context of low efficiency and expensive sequencing.

This inefficiency is also illustrated in the difference between 1-fragment and 4-fragment analysis. Due to the low number of reads/bait, a single fragment only has enough information for the more common, closer interactions. The study therefore requires the pooling of 4-fragments (~12,000

reads, and ~1.6kb resolution) to have enough reads for longer interactions to be called. I think this should be explained more explicitly in the text, that the single fragment and pooled fragments analysis give different interactions because single fragments have insufficient sequence depth to call longer range interactions, therefore a lower resolution, 4 fragment analysis had to be employed for these interactions. I don't think the mechanism for how the 'median distance for cis interactions increased when decreasing the resolution' is obvious to the reader as it is currently written, and again is important for anyone wishing to utilise this method, and understand why this is the case.

Finally the other methods could be used to target promoters of non-coding genes or alternative promoters - it would just mean using different baits! (actually been done in some studies).

Reviewer #2 (Remarks to the Author):

Summary:

Myriads of genome-wide associations studies have shed light on the genetic loci influencing the susceptibility to a variety of complex disorders or quantitative traits. However, only about 7% of the associated SNPs reside in the coding genome. To make sense of the SNPs associated with bone mineral density (BMD) located in the non-coding genome Chesni and colleagues performed a 3C experiment combined with capturing. They used custom designed capture probes for 95% of all known promoters resulting in about 38.000 viewpoints. In parallel, RNAseq, ATACseq, and ChIPseq for several histone marks were performed. Comparing the resulting interaction peaks with 110 GWAS hits for BMD they found meaningful interactions with promoters for 33 (roughly 30%). Three such loci are presented and investigated in more detail: 1. SMAD3, 2. WNT16, 3. SFRP4. Interestingly, the main looping of the region harboring a SNP previously assigned to WNT16 occurred with the promoters of the genes ING3 and CPED1. Only knockdown of ING3 resulted in lower alkaline phosphatase and mineralization levels after osteogenic stimulation of MSCs. Likewise, in the SFRP4 region SFRP4 and EDR1 were knocked down, with the latter showing a tendency towards reduced mineralization.

Major Comments:

The subject of the paper is of high relevance for the field of bone and osteoporosis research. The research question is clearly delineated in the introduction and the methods are comprehensively described. Looking at the data there is no doubt that the experiments have been carefully planned and successfully carried out.

However, it is difficult to understand why the authors chose the 110 GWAS loci identified by Kemp et al. in 2014, although the same first author published a more recent study describing 203 loci in 2017. The analysis should be performed again for the 203 GWAS hits described by Kemp et al. 2017.

The study design has several disadvantages: 1. Although 1.6 billion reads were produced per sample the immense number of viewpoints has the effect that the detected interactions are only based on a few reads. 2. Since the capture probes are designed in house the dataset cannot be compared to similar studies that have been conducted e.g. by the Epigenome Consortium. A criterion for the relevance of promoter-enhancer interactions is their cell type specificity under the assumption that a specific phenotype is more readily explained by an interaction found in a relevant cell type and not in any cell type of the human body. It would therefore be advantageous to see parallel genome-wide results from another cell type generated with the SPATIALseq approach. At least for the three highlighted candidate loci the chromatin looping should be analyzed in MSCs differentiated into another cell type (chondrocyte or adipocyte, see below).

Although the authors generate so much data it is surprising that so little of this information is presented in the main figures. Moreover, it is unfortunately not possible to get access to processed data via the mentioned link. The connecting arches shown in Figures 1 to 3 do not allow to judge the validity of the promoter-enhancer interactions. For all 33 loci mentioned in Table 1 supplemental figures should be given including the tracks for SPATIALseq, ATACseq, H3K27Ac ChIPseq.

If SPATIALseq tracks provide a too low coverage to be displayed as a track, the interaction profiles in the important figures 1 to 3 should be reproduced by independent 4C experiments in MSCs differentiated into osteoblasts and adipocytes, respectively. Besides ATACseq and H3K27Ac ChIPseq also the TAD boundaries should be given for these three loci.

The knockdown experiments shown in Figure 3 and 4 are not entirely convincing. Error bars for expression analysis are very high. While in Figure 3 after knockdown of ING3 at least a slight upregulation of ID1 and RUNX2 and a reduction of ALPL expression was found the only effect in Figure 4 is the reduced alizarin red staining after EPDR1 knockdown. However, ALP and alizarin red stainings are not quantified. It would strengthen the conclusions if these readouts would be measured in at least three experiments and statistically analyzed.

Minor Comments:

Line 46: Osteoporosis is not only loss of BMD, but also caused changes in bone architecture and bone material properties finally leading to an elevated fracture rate.

Table 1: It is surprising that the region containing the SNPs rs1107747 and rs1107748 upstream of the SOST gene is not interacting with the SOST promoter. This region is overlapping with the ERC5 region that has been shown to regulate SOST expression. There is even a mouse knockout for this region showing a hyperostosis phenotype like van Buchem disease clearly indicating the enhancer function. The authors should comment on this contradiction.

Reviewer #3 (Remarks to the Author):

In their current manuscript, the authors use a “SPATIaL-seq” approach to determine the relevance of previously identified genetic variants associated with low BMD and thereby identify putative genes involved in the regulation of osteoblast differentiation and BMD. Using human MSC, they demonstrate that siRNA-mediated knockdown of two of them (ING3 and EPDR1) results in impaired osteoblast differentiation. The data presented is interesting and the scientific approach seems to be innovative, although my expertise lies neither in GWAS nor in promoter studies.

Major points:

- 1) From a technical point of view, it would be important to demonstrate efficacy of siRNA knockdown (at least for ING3 and EPDR1) on the protein level.
- 2) From their current data it is not clear whether ING3 and EPDR1 are specifically involved in osteoblast differentiation or simply represent essential genes for e.g. general cell metabolism, cell division or DNA replication. Especially the fact that they do not observe a clear influence on the expression of key osteoblast genes such as *Osx* points towards this direction. It would be therefore important to investigate the impact of the individual siRNA knock downs in MSC during differentiation of e.g. adipocytes and determine whether this differentiation is impaired as well.

POINT-BY-POINT RESPONSES TO COMMENTS

We thank the reviewers for their constructive comments and are grateful for the opportunity to address them and to submit a revised manuscript. Point-by-point responses are presented below.

REVIEWER #1

The paper has used promoter Capture-C, ATAC-seq and RNA-seq on a novel, primary cell type (osteoblasts) for a BMD focused analysis. They demonstrate very nicely the utility of this study design, and show how it can be used to direct GWAS findings to novel genes, not necessary the ones implicated by distance/previous knowledge. As such I feel it could be an important and relevant piece of work, not only to this field but to a wider 'GWAS' community. The work to "validate" the implicated genes is good, using siRNA to knock out novel genes and show they have a potential affect in traits relevant to BMD.

We thank the reviewer for the positive assessment of our manuscript, and are delighted that the importance of this work is recognized.

One of the claims in the paper is to have developed a new methodology. I feel the methodology is Capture C, enriched with Agilent baits designed for promoters - so Capture C. I'm not sure it deserves a new acronym.

We agree with the reviewer that our approach is simply an extension, albeit a sizeable one, on the existing Capture C methodology and therefore, as suggested, we have now removed the 'SPATiaL-seq' acronym completely.

Some of the results section feel more like introduction/methods, for example lines 105 to 110.

As suggested, we have moved the indicated parts to the Introduction.

The major issue is with the use of GWAS data. GWAS SNPs are called 'independent' if the $r^2 < 0.2$ and proxy if the $r^2 > 0.4$. There are now robust, statistical methods to define independent affects in a GWAS region, conditioning on the lead variant.

More troubling is 'proxy SNPs' of $r^2 > 0.4$. This type of functional analysis of GWAS loci should provide robust evidence of causality on robust GWAS hits. I'm not sure any publications use

r²>0.4 to define proxy SNPs for GWAS? More usually it is r²>0.8, and preferably 0.9. Using r²>0.4, I feel, undermines the original GWAS results, such that the functional data is left annotating SNPs that are 'not really associated with BMD'. The 14,007 BMD SNPs, that are used to annotate ones linked to genes, are likely to be a massive overestimate, and the analysis should be performed on a more realistic (r²>0.8) set of SNPs, to provide SNPs that genuinely have a chance of being causal, and are linked to genes.

Firstly, we have now added the analysis of 307 conditionally independent SNPs from 203 loci (from Kemp et al., Nat Genet 2017), as also suggested by reviewer #2. Secondly, and very crucially, the reviewer makes an excellent point regarding the r² threshold's choice, which is somewhat subjective and something we have indeed debated internally. Our idea was to be as inclusive as possible in our analyses, so not to lose any interesting lead (indeed, for the candidate SNPs at the *CPED1* locus influencing *ING3*, r² with the sentinel SNP was ~ 0.6). As an example, a very well-established interaction between a bone distal enhancer region containing SNPs rs1107747 and rs1107748 (r² ~ 0.7 with the sentinel GWAS SNP) and the *SOST* promoter (see also the comment from reviewer 2) would have been missed if we had used the more restrictive 0.8 threshold. However, we recognize that an r² threshold of 0.8 gives more confidence that the analyzed variants are indeed associated with the GWAS signals – and indeed much more in line with what one might expect from a credible set analysis. Therefore, given this important guidance from the reviewer, we now present the analysis with the 0.8 threshold in the main text, but continue to include the results for the much more liberal 0.4 threshold in the Supplementary Information.

The authors compare their method to other published methods (I think its wrong to imply HiChIP is 'biased' - it asks a different question to this study design, for example about how active enhancers interact with promoters. Maybe hypothesis driven?).

As suggested, and in agreement with the reviewer, we have changed the text in the Discussion to "hypothesis-driven nature of the Hi-ChIP technique".

I feel the inefficiencies in this methodology are being compensated for by a large sequencing effort. There is a wealth of techniques available for this work now, and I think its important both the strengths and limitations are clearer for people to make an informed decision of which study design to select. For example, typical of Capture C this study produces around 5% of 'usable' data (Unique di-tags) from the original sequencing reads, which compares unfavourable with other methods (for example Javierre et al Cell 2016 - promoter capture HiC had around 25% unique di-tags). For this type of interaction data typically 10,000 di-tags /bait are optimal (Javierre had ~20,000). Here it is around 3,000/bait (hence the requirement for pooling to call longer interactions). So even though they employed the massive

throughput (and expensive) NovaSeq, the study still produced a limited number of unique di-tags per fragment. There may well be advantages of this methodology - lower cell number?, quicker?, but these need to be explained in the context of low efficiency and expensive sequencing.

We thank the reviewer for this comment, and we have expanded on these points in the Discussion accordingly, by stating “Typical of Capture C, our study produced around 5% of 'usable' data (unique di-tags) from the original sequencing reads, which compares unfavourably with other methods such as promoter capture HiC which typically yields approximately 25% unique di-tags. However, given the fact that the approach we employed yields much higher resolution, we believe the relative high sequencing cost returned significantly more information”.

This inefficiency is also illustrated in the difference between 1-fragment and 4-fragment analysis. Due to the low number of reads/bait, a single fragment only has enough information for the more common, closer interactions. The study therefore requires the pooling of 4-fragments (~12,000 reads, and ~1.6kb resolution) to have enough reads for longer interactions to be called. I think this should be explained more explicitly in the text, that the single fragment and pooled fragments analysis give different interactions because single fragments have insufficient sequence depth to call longer range interactions, therefore a lower resolution, 4 fragment analysis had to be employed for these interactions. I don't think the mechanism for how the 'median distance for cis interactions increased when decreasing the resolution' is obvious to the reader as it is currently written, and again is important for anyone wishing to utilise this method, and understand why this is the case. Finally the other methods could be used to target promoters of non-coding genes or alternative promoters - it would just mean using different baits! (actually been done in some studies).

We fully agree with the points the reviewer makes. In order to clarify, we changed the text in the Results section about the 1-fragment and 4-fragment analyses to: "We first performed an analysis at 1-fragment resolution and observed that single fragments had insufficient sequence depth to call longer range interactions. We therefore performed an analysis at a lower resolution (4-fragment windows, mean size ~1736 bp; median 1440 bp) to increase sensitivity at longer interaction distances". We then merged the results from the two analyses, taking advantage of a very high resolution for short-distance interactions and trading off resolution, albeit still double the resolution of typical promoter capture HiC using HindIII for increased sensitivity at longer interaction distances."

We go on to state that “While other comparable methods can also be scaled to target promoters of non-coding genes or alternative promoters, combined with the high resolution we achieve, this approach returns extremely informative data”.

REVIEWER #2

Myriads of genome-wide associations studies have shed light on the genetic loci influencing the susceptibility to a variety of complex disorders or quantitative traits. However, only about 7% of the associated SNPs reside in the coding genome. To make sense of the SNPs associated with bone mineral density (BMD) located in the non-coding genome Chesi and colleagues performed a 3C experiment combined with capturing. They used custom designed capture probes for 95% of all known promoters resulting in about 38,000 viewpoints. In parallel, RNAseq, ATACseq, and ChIPseq for several histone marks were performed. Comparing the resulting interaction peaks with 110 GWAS hits for BMD they found meaningful interactions with promoters for 33 (roughly 30%). Three such loci are presented and investigated in more detail: 1. SMAD3, 2. WNT16, 3. SFRP4. Interestingly, the main looping of the region harboring a SNP previously assigned to WNT16 occurred with the promoters of the genes ING3 and CPED1. Only knockdown of ING3 resulted in lower alkaline phosphatase and mineralization levels after osteogenic stimulation of MSCs. Likewise, in the SFRP4 region SFRP4 and EDR1 were knocked down, with the latter showing a tendency towards reduced mineralization.

Major Comments:

The subject of the paper is of high relevance for the field of bone and osteoporosis research. The research question is clearly delineated in the introduction and the methods are comprehensively described. Looking at the data there is no doubt that the experiments have been carefully planned and successfully carried out.

We very much appreciate the positive comments regarding our manuscript. However, we must point out to the reviewer that our study did not actually involve the specific generation of ChIPseq data, as we used the ENCODE resource for the latter.

However, it is difficult to understand why the authors chose the 110 GWAS loci identified by Kemp et al. in 2014, although the same first author published a more recent study describing 203 loci in 2017. The analysis should be performed again for the 203 GWAS hits described by Kemp et al. 2017.

We initially chose the 2014 BMD loci because they were determined by DEXA, the standard tool used to assess BMD. However we recognize that the overlap between the DEXA and the heel ultrasound BMD signals is very substantial, so we agree with the reviewer and have now expanded our analysis to include the BMD heel ultrasound loci from Kemp et al.

The study design has several disadvantages: 1. Although 1.6 billion reads were produced per sample the immense number of viewpoints has the effect that the detected interactions are only based on a few reads. 2. Since the capture probes are designed in house the dataset cannot be compared to similar studies that have been conducted e.g. by the Epigenome Consortium. A criterion for the relevance of promoter-enhancer interactions is their cell type specificity under the assumption that a specific phenotype is more readily explained by an interaction found in a relevant cell type and not in any cell type of the human body. It would therefore be advantageous to see parallel genome-wide results from another cell type generated with the SPATIALseq approach.

We acknowledge the inefficiencies pointed out by both reviewers 1 and 2, and we have expanded on these points in the Discussion accordingly, by stating “Typical of Capture C, our study produced around 5% of 'usable' data (unique di-tags) from the original sequencing reads, which compares unfavourably with other methods such as promoter capture HiC which typically yields approximately 25% unique di-tags. However, given the fact that the approach we employed yields much higher resolution, we believe the relative high sequencing cost returned significantly more information”.

To address cell type specificity and relevance of the SNP-promoter interactions, we compared the osteoblast results with HepG2 cell line data we had available. From a global perspective, only two loci revealed the same interactions between the two cell types.

At least for the three highlighted candidate loci the chromatin looping should be analyzed in MSCs differentiated into another cell type (chondrocyte or adipocyte, see below).

We now present data from MSC-derived adipocytes for the three specific loci. Interactions at the *CPED1* and *STARD3NL* loci are present in the MSC-derived adipocytes, while no interactions are detected at *SMAD3*. However, given the observations we outline below, this may not be entirely surprising.

Although the authors generate so much data it is surprising that so little of this information is presented in the main figures. Moreover, it is unfortunately not possible to get access to processed data via the mentioned link. The connecting arches shown in Figures 1 to 3 do not allow to judge the validity of the promoter-enhancer interactions. For all 33 loci mentioned in Table 1 supplemental figures should be given including the tracks for SPATIALseq, ATACseq, H3K27Ac ChIPseq.

We agree with the reviewer that processed data should be made available and we apologize for this oversight. Because of our expanded analysis (DEXA plus heel ultrasound BMD loci), there

are now 46 GWAS loci corresponding to 77 baited fragments in Table 1 that fulfilled the Capture-C and ATAC-seq constraints we placed on the data. Since a Capture-C track is required for each bait, presenting all the data would result in 46 supplemental figures with 77 tracks. To overcome this, we now provide interaction data in CHiCAGO interaction format, plus the HiCUP pre-processed bam file containing the aligned valid reads from which bigWig coverage tracks can be obtained for any desired bait (through the script now provided in the Supplementary Materials). For ATAC-seq, peak calls in bed format and coverage tracks in bigWig format are now also provided. Please note that, as mentioned above, this study does not involve generation of ChIPseq data. We added a Supplementary Figure (S3) for the three main loci pursued, including tracks for Capture-C and ATAC-seq, plus the ENCODE generated H3K27ac ChIPseq track (GEO accession GSM733739).

Our data are available from ArrayExpress (<https://www.ebi.ac.uk/arrayexpress>) with the following credentials:

Capture-C: MSC-derived osteoblasts, E-MTAB-6862, username Reviewer_E-MTAB-6862, password r7rvE6v0; HepG2: E-MTAB-7144, username Reviewer_E-MTAB-7144, password voeeyYy.

ATAC-seq: MSC-derived osteoblasts, E-MTAB-6862, username Reviewer_E-MTAB-6834, password 9hheoywn; HepG2; E-MTAB-6835, username XXX, password XXX.

RNA-seq: MSC-derived osteoblasts, E-MTAB-6835, username Reviewer_E-MTAB-6835, password ywehxxx3.

Data are currently under curation and we are waiting for the reviewer's credentials for the HepG2 ATAC-seq dataset. We will email them to the editor as soon as they are available.

If SPATIALseq tracks provide a too low coverage to be displayed as a track, the interaction profiles in the important figures 1 to 3 should be reproduced by independent 4C experiments in MSCs differentiated into osteoblasts and adipocytes, respectively. Besides ATACseq and H3K27Ac ChIPseq also the TAD boundaries should be given for these three loci.

TAD boundaries are now included (Supplementary Figure S3). Given that all our Capture-C experiments were conducted in triplicate, we feel that validation through 4C, which is a low-throughput and time-consuming method that is not sufficiently dissimilar to Capture-C, is not required. We instead focused our efforts on further functional characterization of the novel genes identified.

The knockdown experiments shown in Figure 3 and 4 are not entirely convincing. Error bars for expression analysis are very high. While in Figure 3 after knockdown of ING3 at least a slight upregulation of ID1 and RUNX2 and a reduction of ALPL expression was found the only

effect in Figure 4 is the reduced alizarin red staining after EPDR1 knockdown. However, ALP and alizarin red stainings are not quantified. It would strengthen the conclusions if these readouts would be measured in at least three experiments and statistically analyzed.

The gene expression analysis after knockdown experiments are averaged from MSCs that were harvested from at least three completely independent donors. There is known significant patient to patient heterogeneity in MSC behavior, as one would expect across the human population. Since the change in gene expression (especially Id1, ALP, and Osterix) differed considerably across donor lines, our standard deviation is high. As an example to show the results within a single donor, we have posted the results below. If required editorially, we can provide the within-donor data for each donor utilized.

We have also quantified the ALP and alizarin red staining using image analysis from three individual donor lines (at least five separate experiments for ALP and four separate experiments for alizarin red staining). Statistical analyses were also performed, showing that *ING3* and *EPDR1* targeting significantly reduced both the ALP and alizarin red staining whereas *CPED1*, *WNT16* or *SFRP4* targeting did not significantly change the amount of ALP and Alizarin red S. This new data is shown in Figures 2 and 3.

Minor Comments:

Line 46: Osteoporosis is not only loss of BMD, but also caused changes in bone architecture and bone material properties finally leading to an elevated fracture rate.

We have added new wording according to the reviewer's suggestion.

Table 1: It is surprising that the region containing the SNPs rs1107747 and rs1107748 upstream of the SOST gene is not interacting with the SOST promoter. This region is overlapping with the ERC5 region that has been shown to regulate SOST expression. There is even a mouse knockout for this region showing a hyperostosis phenotype like van Buchem

disease clearly indicating the enhancer function. The authors should comment on this contradiction.

We thank the reviewer for catching this. It was indeed a typo in Table 1, now updated to show the interaction between rs1107747 and rs1107748 and SOST.

REVIEWER #3

In their current manuscript, the authors use a “SPATIAL-seq” approach to determine the relevance of previously identified genetic variants associated with low BMD and thereby identify putative genes involved in the regulation of osteoblast differentiation and BMD. Using human MSC, they demonstrate that siRNA-mediated knockdown of two of them (ING3 and EPDR1) results in impaired osteoblast differentiation. The data presented is interesting and the scientific approach seems to be innovative, although my expertise lies neither in GWAS nor in promoter studies.

We very much appreciate the time the reviewer took to assess our manuscript.

Major points:

1) From a technical point of view, it would be important to demonstrate efficacy of siRNA knockdown (at least for ING3 and EPDR1) on the protein level.

We have now added immunoblot data to show protein levels of *ING3* and *EPDR1* after siRNA knockdown. Each siRNA effectively reduced protein levels of its corresponding target: *ING3* (~70% knockdown) and *EPDR1* (>85% knockdown) as quantified from four independent hMSC donor lines. Additionally, SMAD phosphorylation was also detected to evaluate the effect of *ING3* and *EPDR1* silencing on canonical BMP signaling. Our results showed that, although *ING3* knockdown did not affect canonical BMP signaling, *EPDR1* silencing slightly reduced SMAD phosphorylation. The band intensities were calculated and averaged across donor and this new data is shown in the new Figure 4.

2) From their current data it is not clear whether ING3 and EPDR1 are specifically involved in osteoblast differentiation or simply represent essential genes for e.g. general cell metabolism, cell division or DNA replication. Especially the fact that they do not observe a clear influence on the expression of key osteoblast genes such as Osx points towards this direction. It would be therefore important to investigate the impact of the individual siRNA knock downs in MSC

during differentiation of e.g. adipocytes and determine whether this differentiation is impaired as well.

This was a very interesting question, particularly considering the reciprocal association between osteoblastogenesis and adipogenesis. As requested by the reviewer, we performed siRNA knockdown of *ING3* and *EPDR1*, and subjected the targeted cells to adipogenic differentiation. Unlike osteogenic differentiation, remarkably, silencing of each of these genes enhanced adipogenic differentiation as demonstrated by significant accumulation of lipid droplets over 21 days (two independent donor lines) as well as enhanced expression of C/EBP alpha (across three independent hMSC donor lines). These results now shown in the new Figure 5 suggest that both *ING3* and *EPDR1* are effector genes modulating the fate of hMSC differentiation, rather than just having an indirect effect. These results are also consistent with the fact that we see consistent contacts for these two loci between the MSC derived osteobalsts and adipocytes.

Reviewer #1 {had no further comments to the authors}

Reviewer #2 (Remarks to the Author):

We thank the authors for their efforts to answer all the points raised by the reviewers and to even include additional functional experiments. This has increased the quality of the manuscript. However, there are still some concerns.

1. Additional profiles provided in the supplement for the loci SMAD3, CPED1, and EPDR1 were provided. However, in the current form these data are not sufficient to dissolve this reviewer's doubt. The peaks adjacent to the bait probes are so high that the peaks at longer distance can hardly be judged. In fact, there are hardly any peaks at all. At the SMAD3 locus only one of the roughly dozen long-range interactions indicated by the spider plots can clearly be associated with one Capture C peak overlapping an ATAC-seq peak. Usually, H3K27Ac peaks are additionally considered for defining an enhancer, but these cannot be judged since the GSM733739 track provided only gives a continuous block around the gene. Thus, it is still possible that the low coverage per locus (roughly 2800 reads per target, other studies like Javierre et al. used 6800 reads, or Andrey et al. even almost 13000) does not allow for a reliable peak calling.

In order to make the data more convincing the high peaks in the profiles adjacent to the bait should be removed as they are not relevant for the interpretation of the data. Usually 5-10kb around the baits can be removed. This would allow to really appreciate the number of reads per region and the height of the peaks. Only the more meaningful 4-fragment data should be presented together with ATAC-seq, the individual H3K27Ac peaks, and the spider plots so that the co-localization of the Capture C peaks and the called interactions can be regarded. This should be done for all genes discussed in the paper as well as the well-known BMD-associated genes SOST and EN1 with SNPs with gene regulatory effect.

2. Unfortunately, yet another GWAS study incorporating heel ultrasound data from even more probands has been published (Morris et al. 2018). These authors also used ATAC-seq in osteoblasts to define the relevant SNPs resulting in 1103 independent SNPs influencing BMD. Although the genome-wide Capture C approach used here leads to lower read depth and problems with peak calling the immense advantage is that the additional SNPs found in this large study can be analyzed in the existing dataset. It would further enhance the impact of this study and might lead to interesting additional discoveries.

3. The data in Figures 2-5 showing the effects on knockdown of the candidate genes on osteoblast and adipocyte differentiation should be reduced and made more comprehensive. In the current form they dominate the paper, which is not really justified. I would suggest to merge Figures 2 and 3. The prove for the successful knockdown and the Osx (correct gene name is SP7) expression data can be put into the supplementary data so that only the expression of ID1, RUNX2, ALPL, and the functional readouts ALP and alizarin red are given for all five genes analyzed.

Reviewer #3 (Remarks to the Author):

The authors sufficiently addressed my concerns and from my Point of view the manuscript is now acceptable for publication.

RESPONSE TO REVIEWER #2:

“We thank the authors for their efforts to answer all the points raised by the reviewers and to even include additional functional experiments. This has increased the quality of the manuscript”...

We thank the reviewer for recognizing our efforts and acknowledging the increased quality of the manuscript.

“1. Additional profiles provided in the supplement for the loci SMAD3, CPED1, and EPDR1 were provided. However, in the current form these data are not sufficient to dissolve this reviewer’s doubt. The peaks adjacent to the bait probes are so high that the peaks at longer distance can hardly be judged. In fact, there are hardly any peaks at all. At the SMAD3 locus only one of the roughly dozen long-range interactions indicated by the spider plots can clearly be associated with one Capture C peak overlapping an ATAC-seq peak. Usually, H3K27Ac peaks are additionally considered for defining an enhancer, but these cannot be judged since the GSM733739 track provided only gives a continuous block around the gene. Thus, it is still possible that the low coverage per locus (roughly 2800 reads per target, other studies like Javierre et al. used 6800 reads, or Andrey et al. even almost 13000) does not allow for a reliable peak calling.

In order to make the data more convincing the high peaks in the profiles adjacent to the bait should be removed as they are not relevant for the interpretation of the data. Usually 5-10kb around the baits can be removed. This would allow to really appreciate the number of reads per region and the height of the peaks. Only the more meaningful 4-fragment data should be presented together with ATAC-seq, the individual H3K27Ac peaks, and the spider plots so that the co-localization of the Capture C peaks and the called interactions can be regarded. This should be done for all genes discussed in the paper as well as the well-known BMD-associated genes SOST and EN1 with SNPs with gene regulatory effect”.

We agree with the reviewer that it is challenging to call interaction peaks by eye using the Capture C coverage tracks, even when removing the high-coverage bait regions [as suggested, we now provide new figures (Fig. S3) with 5-10kb around the bait region removed, plus the H3K27ac peak track]. However, we would like to clarify what was carried out in our analysis. CHiCAGO is not a peak caller, like HOMER or MACS2, but rather was developed specifically for high-throughput Capture C. It uses a sophisticated background model accounting for ‘Brownian collisions’ (distance-dependent) plus ‘technical noise’, and borrows information across all interactions. Significant interactions are called using a p-value weighting procedure based on the expected true positive rates at different distances, accounting for multiple testing correction. For these reasons, we believe CHiCAGO does achieve higher sensitivity than what is possible by just looking at the coverage tracks by eye in search of evident peaks, or with a peak caller algorithm. Besides, for the five loci analyzed more in depth in the manuscript, the number of valid reads was on average (at 4-fragment resolution) ~9700 reads per bait, which compares favorably with the other studies cited by the reviewer. However, it is our position that orthogonal proof of the validity of the interactions detected by Capture C is the best way to go, and we made substantial efforts to validate some of the identified putative targets with functional

experiments in relevant model systems. Finally, we now provide (Fig. S4) plots of the raw read counts vs distance from bait, with the CHiCAGO-generated expected counts and their 95% confidence intervals, for the five key loci plus *SOST* (significant interactions are highlighted in red). These figure should be a more useful way to display the significant interactions than the coverage tracks.

“2. Unfortunately, yet another GWAS study incorporating heel ultrasound data from even more probands has been published (Morris et al. 2018). These authors also used ATAC-seq in osteoblasts to define the relevant SNPs resulting in 1103 independent SNPs influencing BMD. Although the genome-wide Capture C approach used here leads to lower read depth and problems with peak calling the immense advantage is that the additional SNPs found in this large study can be analyzed in the existing dataset. It would further enhance the impact of this study and might lead to interesting additional discoveries”.

We agree that analysis of the newest heel ultrasound loci in our dataset will provide additional interesting leads, but is beyond the scope of the current manuscript.

3. The data in Figures 2-5 showing the effects on knockdown of the candidate genes on osteoblast and adipocyte differentiation should be reduced and made more comprehensive. In the current form they dominate the paper, which is not really justified. I would suggest to merge Figures 2 and 3. The prove for the successful knockdown and the *Osx* (correct gene name is *SP7*) expression data can be put into the supplementary data so that only the expression of *ID1*, *RUNX2*, *ALPL*, and the functional readouts *ALP* and alizarin red are given for all five genes analyzed.

As suggested, we have merged Figures 2 and 3 and moved both the *SP7* expression data and the proof of successful knockdown to the Supplementary Material.

Reviewer #2 (Remarks to the Author):

All queries have been addressed. No further comments.